# Assessment of the implementation fidelity of a strategy to scale up integrated care in five European regions: a multimethod study

Liset Grooten [1], Isabelle Natalina Fabbricotti,[2] Dirk Devroey [1], Hubertus J M Vrijhoef[3,4]

¹Department of Family Medicine and Chronic Care, Vrije Universiteit Brussel, Brussel, Belgium
²Erasmus School of Health Policy & Management, Erasmus University Rotterdam, Rotterdam, The Netherlands
³Department of Patient & Care, Maastricht Universitair Medisch Centrum+, Maastricht, The Netherlands
⁴Panaxea B.V, Amsterdam, The Netherlands

**Correspondence to**
Ms Liset Grooten;
fennechien.grooten@vub.be

## ABSTRACT

**Objective** The SCaling IntegRated Care in COntext (SCIROCCO) project tested a step-based scaling up strategy to explore what and how to scale up integrated care initiatives in five European regions. To gain a profound understanding of which factors influence the implementation of this strategy, the objective of this study was to assess the extent to which the SCIROCCO strategy was implemented as planned.

**Design** Multimethod study.

**Methods** The extended version of the conceptual framework for implementation fidelity was used to evaluate what factors influence the implementation of the scaling up strategy. Data were collected in the five participating European regions during the intervention period. Data collection methods included: key informant interviews, focus groups, questionnaire studies and project documents.

**Results** All three main steps of the scaling up strategy were implemented with acceptable fidelity. Variations were observed in the duration of implementing the steps in the regions. Also, variations were observed in the coverage of experts to participate in the steps of the strategy. Several factors were observed to influence the implementation: facilitation conditions (ie, good coordination for implementation) and participant responsiveness (ie, a positive experience of participants in the organised study visits). Factors that may have moderated adherence to the original plan of the strategy were found in facilitating conditions (ie, in the flexible approach), participant recruitment factors (ie, adaptions of the procedure by the regions) and contextual factors (ie, the level of development of integrated care).

**Conclusion** This was the first study to assess implementation fidelity of a European project that used a step-based scaling up strategy in five European regions. Similar European projects that are based on collaboration between several European regions can learn from the lessons captured in SCIROCCO and can become more aware of the facilitating factors and pitfalls of implementing such projects.

## Strengths and limitations of this study

► This study was the first to assess the extent to which a step-based scaling up strategy in five European regions was implemented as planned within a European project.

► All five components of the scaling up strategy were implemented with acceptable fidelity. The insights obtained could support other regions interested to use the Scaling Integrated Care in Context tool and processes for achieving knowledge transfer and ultimately scale up their integrated care initiatives.

► Direct involvement as researchers in this European Union (EU) funded project provided a unique opportunity to follow the project closely and to obtain solid cooperation to conduct the research activities. However, choices needed to be made regarding the data collection to align the research activities within the larger EU project and to be feasible to undertake it within the project time.

## BACKGROUND

Numerous integrated care (IC) initiatives have been developed around the world to bring about accessible, high-quality, effective and sustainable health and social care.[1] Consequently, valuable lessons on the design and implementation of the transformation towards IC have been gained.[2] However, less known is what factors contribute to the progress and success of IC initiatives. Obtaining such insights is challenging, which is partially induced by deficient and absent evaluation and measurement of IC initiatives, resulting in the lack of evidence on the working mechanisms in IC.[3] In addition, many of the available assessment tools for IC have been poorly or insufficiently validated.[4] Furthermore, in the presence of innumerable models of IC, there is no universal conceptual understanding of IC, which leads to a lack of understanding on what 'integration' might variously look like. As a result, insight in how, when and why IC initiatives achieve certain results is blurry.[5 6] This lack of knowledge means that it is hard to learn from experiences[7–9] and to know to

what extent and in which form IC initiatives can be implemented in different settings.[10 11]

To explore how to learn from experiences and knowledge of current IC initiatives, the Scaling IntegRated Care in Context (SCIROCCO) project tested a structured approach intended to facilitate upscaling of successful initiatives. This EU subsidised project did so by focusing on the context and environment (ie, the regional delivery system and political and organisational environment) of IC initiatives. A step-based scaling up strategy was implemented as part of the project with the aim to facilitate the implementation of Good Practices (GPs) in IC at local, regional or country level by recognising the maturity requirements of GPs and health and care systems in order to achieve scaling up and knowledge transfer among European Member States.[12] The strategy was implemented in five regions (Basque Country (Spain), Norrbotten (Sweden), Puglia (Italy), Olomouc (Czech Republic) and Scotland (UK)).

The SCIROCCO step-based strategy consists of multiple components that are planned to be implemented in five different settings. The implementation of complex and multicomponent interventions rarely happens as planned.[13] A variety of factors can influence the implementation of a strategy (like SCIROCCO's), such as the context in which an intervention is implemented[14] and the complexity of the intervention.[15] The local contexts of the different regions in SCIROCCO might have led to necessary adaptations during the implementation of the scaling up strategy, and these adaptations might influence the outputs of the strategy. One way to evaluate the implementation of the strategy is by assessing the implementation fidelity. Implementation fidelity refers to 'the degree to which…programs are implemented… as intended by the program developers'.[13] This study, therefore, assessed the fidelity of the implementation of the step-based strategy to examine how the SCIROCCO strategy was delivered in practice.

The approach we used for the assessment of implementation fidelity is based on the work of Carroll et al[16] and Hasson.[17] The measurement of implementation fidelity is a measurement of adherence, with its subcategories content, frequency, duration and coverage (dose).[16] Several moderating factors are suggested in the framework to influence the level of fidelity. These factors are: participant responsiveness; programme complexity; comprehensiveness of policy description; strategies to facilitate implementation; quality of delivery; recruitment; and context. In this paper, we present findings about the extent to which 'the activities within the SCIROCCO project have been implemented in line with expectations and if, how, and how far relevant initiatives have been developed between 2016 and 2018'.[18] In this way, factors that facilitated the implementation and factors that were not conducive to the implementation of the strategy are revealed. These insights in the implementation and contextual conditions of the SCIROCCO strategy will provide lessons for those interested in

SCIROCCO's step-based approach and future use of its tool and processes to reach progress in the integration of care services. In addition, we aim to contribute to a better understanding on what works in IC initiatives and how successful ones can be scaled up.

## METHODS
### SCIROCCO's scaling up strategy
The SCIROCCO consortium was responsible for the implementation of the strategy. The implementation of the step-based scaling up strategy in the five participating regions was the specific responsibility of the regional partners in each of the five regions. Three regional partners were lead partners for the development of one of the three steps of the scaling up strategy (in collaboration with the other partners within the project). The other partners (referred to as supportive partners) were responsible for the supportive activities within the wider project. These included project coordination, dissemination activities, development of the tool, capturing lessons on the strategy and one independent evaluator (more details are provided in our study protocol). The plan and details on how to implement the step-based strategy were described in the grant agreement (protocol). However, not all exact details have been provided as it was indicated that the exact outline and content of several activities were to be developed during implementation. The description of the step-based strategy, as provided in the protocol, is briefly described below.

### Step 1: assessment of maturity requirements in selected GPs
The first step of the SCIROCCO strategy consists of the identification of 30 potentially adoptable GPs in the five regions. Within SCIROCCO Good Practices are defined as real-life examples of successfully applied innovations in IC. The viability of these GPs is assessed using a form that is developed during the implementation. After which 15 GPs are selected as viable scaling up practices. Second, the maturity requirements for transfer of these 15 GPs are assessed using the SCIROCCO tool (see box 1, all details of the tool are presented in online supplementary

> **Box 1    Description of the Scaling IntegRated Care in Context (SCIROCCO) tool**
>
> The SCIROCCO tool consists of 12 dimensions that represent the range of activities that needs to be managed in order to deliver integrated care. The maturity of healthcare system for integrated care or the maturity requirements of Good Practices (GPs) in integrated care are assessed by considering each dimension and allocating a measure of progress or 'maturity' (on a 0–5 scale) to each dimension. The scales include the maturity indicators and reflect the basic indications to look for when assessing the current situation of the maturity of healthcare system for integrated care or the maturity requirements of GPs. After the assessment, a simple graphical representation (ie, spider diagram) of status can be derived that reveals areas of strength, further attention and improvement in each of the 12 dimensions.

Main scaling up steps of SCIROCCO strategy

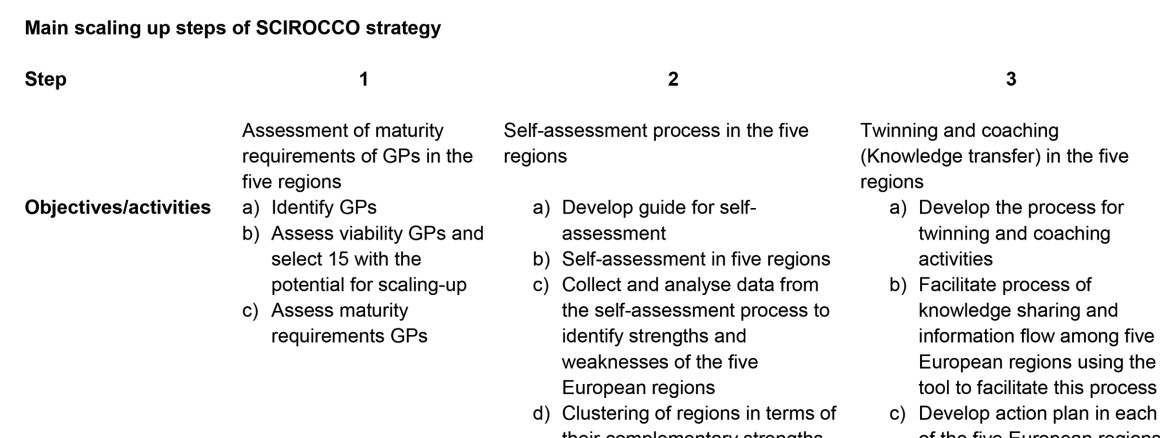

| Step | 1 | 2 | 3 |
|---|---|---|---|
| | Assessment of maturity requirements of GPs in the five regions | Self-assessment process in the five regions | Twinning and coaching (Knowledge transfer) in the five regions |
| Objectives/activities | a) Identify GPs<br>b) Assess viability GPs and select 15 with the potential for scaling-up<br>c) Assess maturity requirements GPs | a) Develop guide for self-assessment<br>b) Self-assessment in five regions<br>c) Collect and analyse data from the self-assessment process to identify strengths and weaknesses of the five European regions<br>d) Clustering of regions in terms of their complementary strengths and weaknesses | a) Develop the process for twinning and coaching activities<br>b) Facilitate process of knowledge sharing and information flow among five European regions using the tool to facilitate this process<br>c) Develop action plan in each of the five European regions |

**Figure 1** Overview of the three main steps and intended activities per step of the SCIROCCO scaling up strategy. GPs, Good Practices; SCIROCCO, Scaling IntegRated Care in Context.

appendix A), resulting in a guide on how to scale up given the local context in which the practices have been developed. An overview of the three main steps and intended activities of the strategy is provided in figure 1.

### Step 2: self-assessment process in the five regions
The second step exists of the development of a guide for the regions describing how to use the tool as a self-assessment tool. Hereafter, the five participating European regions assess their maturity in the adoption for IC, using the online SCIROCCO tool, to identify strengths, gaps and areas for improvement. Subsequently, the outcomes of the assessments are collected and analysed, thereby informing about the maturity gaps of a regional health and care system in IC. The five regions are then clustered in terms of their complementary strengths and weaknesses. Using the outcomes on clustering of regions, the regions are paired in such a way that the knowledge transfer will flow between the regions with the same strengths (twinning) as well as between the regions scoring high at a particular dimension with the regions scoring low along the same dimension (coaching).

### Step 3: knowledge transfer processes in the five regions
The last step consists of the development of the process for the twinning and coaching activities of five European regions to support the creation of twinning and coaching relationships by using the information (steps 1 and 2 of SCIROCCO's strategy) to construct these relationships. The process of knowledge sharing and information flow among the five European regions is facilitated, and one twinning and one coaching activity per region is envisaged. The SCIROCCO tool is used to assist in this process. The twinning and coaching activities are organised as face-to-face meetings, webinars and various other online tools.

The last activity comprises the development of an action plan in each of the five European regions. The action plans reflect the findings of the self-assessment process and specifically focuses on addressing the weaknesses in the maturity of regional health and care systems.

The action plan informs the decision makers about the priority of actions necessary for improvement of their health and care systems. The actual implementation of the plans and monitoring of their progress (ie, ongoing self-assessment) is not within the scope of the implementation, due to the limited duration of the SCIROCCO project.

### Data collection and study population
To assess the fidelity of each scaling-up step, a multi-method approach was used. Data were collected during the implementation of the strategy from each of the partners and local stakeholders involved in implementation of the strategy. Data collection methods included key informant interviews, focus groups, a questionnaire study and the collection of project documents (ie, progress reports and deliverables). For the reporting of study, we used the Consolidated criteria for Reporting Qualitative research.[19]

The extended conceptual framework for implementation fidelity, developed by Hasson[17] and based on the work of Carroll et al,[16] was used for evaluating implementation fidelity of SCIROCCO's step-based approach (the topics are presented in the next section).

The following topics were leading in the data gathering and analysis of the data:
- ► Content (the way in which the steps within the strategy were undertaken and changes in these steps).
- ► Frequency and duration (duration was interpreted by checking the timeline and deadlines for the implementation of the three steps).
- ► Facilitating conditions (quality and usefulness of the protocol, guidance and collaboration within the scaling up strategy).
- ► Context (barriers and facilitators in carrying out the strategy).
- ► Participant recruitment procedures.
- ► Points of improvement.

Qualitative data, including semi-structured interviews and working documents (interim report, final report,

**Table 1** Implementation fidelity components (adherence subcategory and potential moderating factors), research question, data collection procedure/source and planning)

| Adherence subcategory | Research question(s) | Data collection procedures/sources | Measurement planning |
|---|---|---|---|
| Content | How are the three steps of the of scaling up strategy delivered in the five regions? | ▶ Semi-structured interviews with supportive and regional partners.<br>▶ Work documents (ie, progress reports and interim reports) of supportive and regional partners.<br>▶ Deliverables of supportive and regional partners. | Alongside SCIROCCO project. |
| Frequency/duration | How many GP assessments are performed in the five regions? How many self-assessments are performed in the five regions? How many twinning and coaching activities are performed in the regions? What was the duration of the implementation of the three steps? | ▶ Semi-structured interviews with supportive and regional partners.<br>▶ Work documents (ie, progress reports and interim reports) of supportive and regional partners.<br>▶ Deliverables of with supportive and regional partners. | Alongside the SCIROCCO project. |
| Coverage (reach) | How many local stakeholders per region participated in the different steps of the SCIROCCO strategy? | ▶ Work documents of regional partners.<br>▶ Deliverables of regional partners.<br>▶ Emails with regional partners. | Alongside the SCIROCCO project. |
| *Potential moderating factors* | | | |
| Participant responsiveness | How satisfied were the participants with their participation in the study visits? How did the participants perceive the outcomes and relevance of the study visits? | ▶ Focus groups with local stakeholders and regional partners of the five participating regions.<br>▶ Short survey with the local stakeholders and regional partners of the five regions on experience in in the study visit. | After the study visits. |
| Participant recruitment | What recruitment procedures were used to attract local stakeholders to participate in the three steps of the strategy? | ▶ Semi-structured interviews with supportive and regional partners.<br>▶ Focus groups with local stakeholders and regional partners of the five participating regions.<br>▶ Deliverables with supportive and regional partners. | Alongside SCIROCCO project. After the study visits. Alongside SCIROCCO project. |
| Conditions to facilitate implementation | What conditions were used to support the implementation of the SCIROCCO scaling up strategy? How were these conditions perceived by SCIROCCO partners and local stakeholders involved in the strategy? | ▶ Semi-structured interviews with supportive and regional partners.<br>▶ Focus groups with local stakeholders and regional partners of the five participating regions.<br>▶ Work documents (ie, progress reports and interim reports) of with supportive and regional partners.<br>▶ Deliverables of with supportive and regional partners. | Alongside SCIROCCO project. After the study visits. Alongside SCIROCCO project. |
| Context | What factors at political, economical, organisational and work group level affected the implementation of the scaling up strategy? | ▶ Semi-structured interviews with supportive and regional partners.<br>▶ Focus groups with local stakeholders and regional partners of the five participating regions.<br>▶ Work documents (ie, progress reports and interim reports) of supportive and regional partners.<br>▶ Deliverables of supportive and regional partners. | Alongside SCIROCCO project. After the study visits. Alongside SCIROCCO project. |

GPs, Good Practices; SCIROCCO, Scaling IntegRated Care in Context.

project deliverables, local reports and emails), were the main data sources to explore adherence and possible factors that moderate adherence to the implementation of three steps of the SCIROCCO scaling up strategy. In table 1, an overview is provided including the examined implementation fidelity components (adherence subcategory and potential moderating factors) in this study, the accompanied research question, data collection procedure/source and planning.

The semi-structured interviews were conducted over Skype with members of the SCIROCCO project responsible for the different steps in the strategy (n=12). The interviews were held by the first author and lasted about 60 min each. Details on the interviews are presented in online supplementary appendix B.

Furthermore, five focus groups were organised after each of the five twinning and coaching study visits (on location) and included the regional SCIROCCO partners in the five regions and the external local members who participated in the study visits (referred to as 'local stakeholders'). These local stakeholders were recruited by the regional SCIROCCO project partners. The focus groups were alternately facilitated by the first author and two members of work package (WP8), which was part of the SCIROCCO project and focused on collecting lessons learnt on the process of using the SCIROCCO tool and strategy. The three female facilitators possessed a minimum of a master's degree and experience in qualitative research. One of the facilitators holds the position of principal eHealth policy analyst, while the other two were researchers working at universities. At the start of the focus groups, the moderators introduced the focus group and themselves, explaining the purpose of the focus group, and requesting the participants to sign the informed consent form (see ethics statement). All participants received an overview of the focus group questions at the beginning of each study visit. Data collected from these focus groups centred on the experiences of the participants on the study visits that were organised as part of step 3 of the scaling up strategy and included the following topics: content, participant recruitment procedures, participant responsiveness and points of improvement. The focus groups lasted approximately an hour. In table 2, an overview of the characteristics of each focus group is provided. Sometimes the same participants were included in the two focus groups. This happened when participants took part of two study visits. The topics of the focus groups were different. The interviews and focus groups were, after obtaining signed consent, audiotaped and transcribed and during both field notes were made.

After the focus groups were conducted, a survey was distributed among the same participants (n=49) to collect data about participants responsiveness (expectations, satisfaction, clarity and usefulness of results of the study visit). The characteristics of the participants who completed the survey are shown in table 3. Some stakeholders participated in two study visits and completed the survey for both visits.

**Table 3** Characteristics of participants who completed the survey (n=40)

| Overall | No. of participants | |
|---|---|---|
| Region of origin participants | Region 1 (Basque Country)=8. Region 2 (Olomouc)=9. Region 3 (Norrbotten)=10. Region 4 (Puglia)=6. Region 5 (Scotland)=7. | |
| Per study visit: | | |
| Puglia (n=12). | Transferring region*: Puglia=2. | Receiving regions†: Olomouc=5 and Scotland=5. |
| Scotland (n=6). | Transferring region: Scotland=2. | Receiving region: Norrbotten=4. |
| Basque country (n=6). | Transferring region: Basque country=3. | Receiving region: Norrbotten=3. |
| Scotland (n=9). | Transferring region: Scotland=0. | Receiving region: Puglia=4 and Basque Country=5. |
| Norrbotten (n=7). | Transferring region: Norrbotten=3. | Receiving region: Olomouc=4. |

*The transferring region is the region acting as the 'coaching' partner in twinning and coaching activity.
†The receiving region, acts as the ''learning'' partner and is the region seeking support from the transferring region to deploy a Good Practice and/or improve a specific aspect of integrated care.

**Table 2** Characteristics of the focus groups

| Location | Subject of focus group | Participants from region (number) | Total number of participants |
|---|---|---|---|
| 1. Puglia | Experience study visit Puglia on GP in telemonitoring. | Experts from Puglia (3), Olomouc (5) and Scotland (6). | 14 |
| 2. Basque country | Experience study visit Basque country on GP in advanced care planning. | Experts from the Basque country (3) and Norrbotten (3). | 6 |
| 3. Scotland | Experience study visit Scotland on dimension innovation management of the SCIROCCO tool. | Experts from Scotland (3) and Norrbotten (5). | 8 |
| 4. Norrbotten | Experience study visit Norrbotten on dimension eHealth and Information services of the SCIROCCO tool. | Experts from Norrbotten (4) and Olomouc (4). | 8 |
| 5. Scotland | Experience study visit on GP in third sector in Scotland. | Experts from Scotland (3), Puglia (4) and Basque country (6). | 13 |
| | | Grand total: | 49 |

GP, Good Practice; SCIROCCO, Scaling IntegRated Care in Context.

### Patient and public involvement

One patient representative was involved in the twinning and coaching activities of one site and was involved in the focus group.

### Data analysis

Data from the interviews, focus groups and documents were analysed using content analysis.[20] A coding scheme, including the implementation fidelity concepts and each intervention component, was used during the coding process. The scheme was tested independently by two researchers (LG and HJMV) prior to undertaking the coding process. The analysis of the transcripts and documents was conducted in NVivo 12. The first coder (LG) coded all transcripts using the coding scheme, and the second coder (HJMV) operated as a control and coded a random selection of 10% of the transcripts and 10% of the collected documents. The results from this coding process were discussed among the researchers, and any disagreement was resolved until consensus was reached. The surveys collected after the twinning, and coaching study visits were analysed using descriptive methods in SPSS V.25. Validity of the findings was ensured by using member checks (summaries of the interviews were sent to the respondents, numbers were checked by sending confirmatory emails to the responsible partners) and triangulation (ie, data from focus groups, documents and interviews were collected to locate/inform the concepts of implementation fidelity)[21]; this also ensured data saturation.[22]

### RESULTS

In this section, first, the results concerning adherence to the three steps of the scaling up strategy during its implementation in the five regions are presented. Here after, the moderating factors that were found to influence the implementation of the scaling up strategy in the five regions are described.

### Adherence subcategories
#### Content

All the steps of the scaling up strategy (maturity requirements in selected GPs, self-assessment process, and twinning and coaching) were implemented with acceptable fidelity. Also, the methodologies developed during the implementation of the steps, which described how to implement the activities in the regions were followed by the regions to a large extent. More details on the content and any deviations observed during the implementation of the steps in the five regions are described below.

#### Step 1: maturity assessment of GPs

All the five regions collected data on GPs in their region, implemented the viability assessments of selected GPs and assessed the maturity requirements of the prioritised GPs by applying the tool. In order to evaluate the viability for scaling-up of the SCIROCCO GPs, a six-criterion

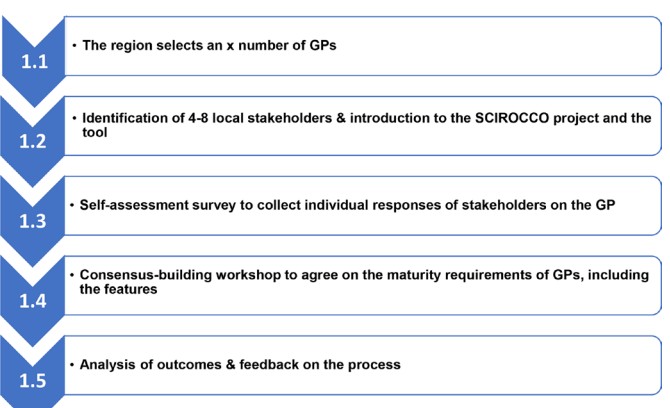

• The region selects an x number of GPs

• Identification of 4-8 local stakeholders & introduction to the SCIROCCO project and the tool

• Self-assessment survey to collect individual responses of stakeholders on the GP

• Consensus-building workshop to agree on the maturity requirements of GPs, including the features

• Analysis of outcomes & feedback on the process

**Figure 2** Revised process on the maturity assessment of Good Practices (GPs). SCIROCCO, Scaling IntegRated Care in Context.

assessment framework was built based on the criteria used in the European Innovation Partnership on Active and Healthy Ageing repository.[23] The six criteria included: (1) time needed for the practice to be deployed; (2) investment per citizen/service user/patient (referring to marginal cost over previous situation); (3) evidence behind the practice; (4) maturity of the practice; (5) estimated time of impact of the practice; and (6) level of transferability of the practice. A deviation was found in delivering the maturity requirement assessment of the selected GPs in the regions; this assessment was performed twice. This was due to observed heterogeneous outcomes across the five SCIROCCO regions. Also, CHAFEA, the fund holder on behalf of Horizon 2020, requested for performing an extra assessment. In figure 2, the process of the revised assessment is presented, and all the five regions followed the steps of the revised process in their region. The main difference between the first and second assessment was the number of assessors and focus of the assessment. In the first assessment, the focus was on the maturity of the context wherein the GP was developed and was performed by a single representative of the GP. In the second assessment, a group of experts assessed the GP with the focus on the maturity needed to implement the GP in different health and social care settings.

#### Step 2: self-assessment in five regions

The five participating regions implemented the self-assessment process according to the developed methodology (as shown in figure 3) and applied the tool. All regions wrote a report on the implemented local self-assessment process in their region, including an analysis on the strengths and weaknesses and priority actions for their region.

#### Step 3: twinning and coaching

Each of the five SCIROCCO regions were involved in the twinning and coaching processes. The process designed to conduct twinning and coaching activities in the regions consisted of three key phases:

Phase 1: planning for the twinning and coaching.

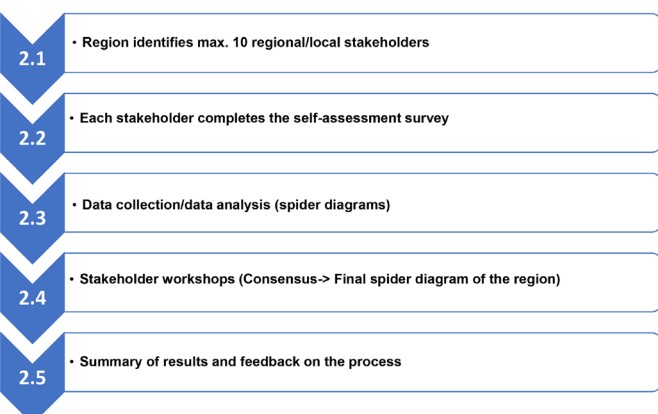

| | |
|---|---|
| **2.1** | • Region identifies max. 10 regional/local stakeholders |
| **2.2** | • Each stakeholder completes the self-assessment survey |
| **2.3** | • Data collection/data analysis (spider diagrams) |
| **2.4** | • Stakeholder workshops (Consensus-> Final spider diagram of the region) |
| **2.5** | • Summary of results and feedback on the process |

**Figure 3** Developed methodology on the maturity assessment of healthcare systems.

Phase 2: knowledge transfer activities.

Phase 3: capturing the outcomes of twinning and coaching.

The twinning and coaching activities took place between a transferring region and receiving region. The transferring region was the region acting as the 'coaching' partner in twinning and coaching activity. The receiving region was the region seeking support and know-how in order to deploy a GP and/or improve a specific aspect of IC and acted as the 'learning' partner. During the implementation of this step, a variation was observed as one region did not play the role of coaching region due to low maturity scoring across all SCIROCCO dimensions. As a result, one region acted twice as transferring region. Another variation was found in the fact that two regions participated twice as receiving region. A description of the GP of one site involved as transferring region in the twinning and coaching activity is provided in box 2. Descriptions of all five twinning and coaching sites are outlined elsewhere.[24]

In figure 4, an overview of the process of phase 2 on the knowledge transfer activity is provided. Local variation in the implementation of these steps per twinning and coaching activity were observed. This was because the individual regions were provided the opportunity to reflect their local needs and strategic priorities for IC in (the scope of) the twinning and coaching process. The implementation of prior contact between the transferring and receiving regions varied (step 3.3). In addition, some of the documentation that were prepared for the twinning and coaching activity were translated. The outline of the programme for the study visits also varied per activity (step 3.4) as the different study visits were adapted to the topic of the visit and the needs of the receiving regions. The general outline of the programme included presentations of the transferring and/or receiving regions, live demonstrations/site visit of the GP (if relevant) and a facilitated discussion on the maturity requirements using the SCIROCCO tool. During the several study visits, the tool was applied in different ways because the partners were interested to test the use of the tool in the twinning and

coaching activity. Furthermore, the tool was not explicitly used in one study visit because of time constraints.

**Box 2    Description of the Good Practice of one site involved as transferring region in the twinning and coaching activities**

Hospital@Home Good Practice is designed as a technological support for already structured activities in home care in Puglia (Italy). The main objective is to reduce hospitalisation and rehospitalisation and to improve the quality of care for patients at home. In addition, the objective is to:

► Reduce the number of patients with heart disease, diabetes and other chronic diseases in the process of instability.
► Activate protected dehospitalisation.
► Optimise the therapy and diagnosis according to international guidelines.

The aim is to implement a new type of telemonitoring, based on continuous collaboration and patient monitoring by different professionals and different users. Patients are telemonitored by their general practitioners by using the innovative home and health monitoring technological solution (H&H Hospital@Home). This solution is able to detect the main clinical and instrumental parameters in addition to the therapeutic administration, based on oxygen and broncoaspiration. It is allocated at the patients' home, and it is permanently interconnected with the general practitioner and/or specialist, by computer, telephone, tablet and other devices.

At the same time, there is a central monitoring room at the hospital for all patients and all devices located at their home. All clinical parameters of patients are stored on a dedicated server, respecting the rules for the respect of privacy. The system allows the healthcare professionals (neurologists, pulmonologist, cardiologists, diabetologists and so on) to monitor and speak with patients remotely. The patients can also activate the visit of the healthcare professionals in their homes. In addition to real-time monitoring of physiological parameters, the healthcare professionals can also monitor the physical and technical characteristics of home device. As a result, it is possible to deliver therapy to the patient remotely.

With regard to the last activity, developing the action plans (phase 3), the developed methodology included the following steps: the receiving regions were asked to organise a local meeting in their region reflect on the outcomes of knowledge transfer activities and agree on

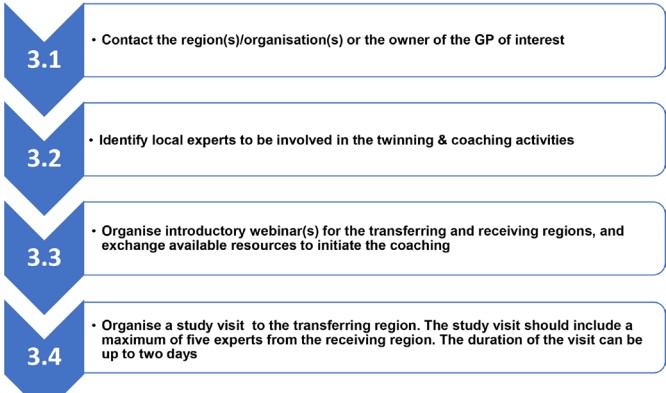

| | |
|---|---|
| **3.1** | • Contact the region(s)/organisation(s) or the owner of the GP of interest |
| **3.2** | • Identify local experts to be involved in the twinning & coaching activities |
| **3.3** | • Organise introductory webinar(s) for the transferring and receiving regions, and exchange available resources to initiate the coaching |
| **3.4** | • Organise a study visit to the transferring region. The study visit should include a maximum of five experts from the receiving region. The duration of the visit can be up to two days |

**Figure 4** Twinning and coaching process. GP, Good Practices.

the local priority actions for the transferability of learning. Here after, the action plans were co-designed by transferring and receiving regions. All five regions participated as receiving regions and wrote an action plan reflecting the outcomes of the twinning and coaching activities. Once completed, the intent was to upload the action plans in the online SCIROCCO tool and share these with all relevant stakeholders. In the last step, the plan was to promote the outcomes of the knowledge transfer activities locally and across the regions; no specific details could be retrieved on the last two intended activities.

### Frequency and duration

In table 4, a general overview is provided on the frequency of the three SCIROCCO steps, to see whether the target that was set for the different steps was met during the implementation of the strategy. The target number of activities were fully executed during implementation, sometimes even exceeded the target. When looking into the revised maturity assessment of GPs (step 2), one deviation was observed during the second GP assessment of one GP in region 5. During the time of the second assessment, the GP was embedded as part of the routine practice at national level; therefore, it was not possible to conduct a second assessment on the GP itself.

During the implementation of the steps, adjustments were made in the timeline of all the steps. An overview of the original timeline and the executed timeline is provided in online supplementary appendix C. A delay was observed in organising the local self-assessment processes in the regions (step 2) since the engagement of the local stakeholders for the self-assessment took much longer than anticipated. In addition, after the self-assessment process, the reassessment of the maturity requirements of 15 GPs (step 1) needed to be executed in the five regions. Since the planning of the implementation of step 3 was dependent on the outcomes of steps 1 and 2, step 3 was organised with a delay of 6 months. The parallel development of SCIROCCO methodology for twinning and coaching allowed the immediate start of the approach in the regions, including the development of regional action plans. In the original programme, when the partners were thinking about the description of the steps, what might have been improved was the duration of the strategy, which was indicated by one supportive partner.

### Coverage

The coverage of the local stakeholders participating in the three main steps of the implemented strategy are shown in table 5. Since the methodology of the different activities was designed during implementation, the target number for identifying the local stakeholders to participate in the different activities per region was not defined beforehand but were indicated during implementation. These target numbers for the different activities are also presented in table 5.

For the revised assessment of step 1, the regional SCIROCCO partners were instructed to identify experts with different profiles, backgrounds and experiences from both managerial and practice profile. The regions adapted the identification of experts to their local context/local GPs, which lead to variation in the structure and the number of stakeholders involved per region. For the activities of step 2 and 3, a variation in structure and the size of the local teams of engaged experts per region was also observed. The local variations were the results of the fact that for step 2, the scope of the self-assessment activity was defined by each individual region reflecting the structure of their healthcare system and the concept of IC and the regions were instructed to capture a diversity of perspectives in the assessment process, including different disciplines, sectors and positions in the organisation. For step 3, the scope of the twinning and coaching process was also defined by each individual region reflecting the local need and strategic priorities for IC and the regional partners selected the stakeholders based on their experiences with the subject of the twinning and coaching activities. This also resulted in variations in the local teams of engaged experts per region.

### Moderating factors
#### Participant recruitment

#### Recruitment for the GP assessment (step 1)
The regional SCIROCCO partners were provided instructions on how to perform the assessments of the GPs in their region and how to select and invite the local stakeholders (GP leaders). The recruitment process for the first assessment of the GPs was implemented in all the five regions in line with the instructions provided. The regional SCIROCCO partners invited the practice leaders to participate in the GP assessment, and they received guidelines that described the steps to follow.

In the second assessment, using the revised methodology, the GP assessment process was changed, and the assessment was performed by a multidisciplinary team rather than single key informant. Again, the regional SCIROCCO partners were introduced to the revised methodology and received new instructions. A slight variation among the regions was observed in the identification procedure of stakeholders. In four regions, the local SCIROCCO project partners identified the local stakeholders themselves (regions 1, 2, 4 and 5). Another region identified local stakeholders with the support of the local steering group (region 3); these stakeholders were included in a local project team that participated in various SCIROCCO activities. All the regional SCIROCCO project partners sent an invitation via email to all the identified local stakeholders in the five regions, and all experts were invited to use the online version of the SCIROCCO tool to conduct their individual assessments. A slight variation was observed as one of the regions organised a meeting to introduce the project to the stakeholders and explain the assessment process (region 1).

**Table 4** Frequency of SCIROCCO's activities (figures represent number unless indicated otherwise)

| Step | Activities | | Project target | Delivered (n (%)) | Region 1(B) | Region 2 (O) | Region 3 (N) | Region 4 (P) | Region 5 (S) |
|---|---|---|---|---|---|---|---|---|---|
| 1 | Number of viability assessment of GPs in the five regions | | 30 | 32 (106.7) | 7 | 4 | 6 | 8 | 7 |
| | Number of GPs selected and assessed for maturity assessment | Assessed with first designed methodology. | 15 | 15 (115.4) | 3 | 3 | 3 | 3 | 3 |
| | | Assessed with revised methodology (including all steps conducted of the revised process). | 15 | 14 (107.7) | 3 | 3 | 3 | 3 | 2 |
| 2 | Regions performed the complete self-assessment process | | 5 | 5 (100) | 1 | 1 | 1 | 1 | 1 |
| | Completed and documented assessments | | 5 | 5 (100) | 1 | 1 | 1 | 1 | 1 |
| 3 | Number of regions that performed role as transferring/coaching region in twinning and coaching activity | | ≥5 | 5 (100) | 1 | 0 | 1 | 1 | 2 |
| | Duration of study visit (up to 2 days) | | | | 1.5 days | 0 day | 1.5 days | 1.5 days | One 1.5 days and one 1 day |
| | Number of regions that performed the role as receiving region | | ≥5 | 7 (140) | 1 | 2 | 2 | 1 | 1 |
| | Agreed action plans to transfer and/or scale up interventions | | ≥5 | 7 (140) | 1 | 2 | 2 | 1 | 1 |

GPs, Good Practices; SCIROCCO, Scaling IntegRated Care in Context.

**Table 5** Coverage of local stakeholders per step of the scaling up strategy

| Step | Activities | Region 1 (Basque Country) | Region 2 (Olomouc) | Region 3 (Norrbotten) | Region 4 (Puglia) | Region 5 (Scotland) | Total |
|---|---|---|---|---|---|---|---|
| 1 | Number of participants assessed the maturity of the context where the GPs was developed (first methodology) | 3×1 | 3×1 | 3×1 | 3×1 | 3×1 | 15 |
| | Number of individual questionnaires collected per GP (4–8 experts maximum) (second methodology) | 1×4 1×4 1×4 | 1×2 1×3 1×3 | 1×5 1×4 1×4 | 1×5 1×5 1×6 | 1×3 1×4 | 56 |
| | Number of participants in face-to-face workshop per GP (second methodology) | 1×4 1×4 1×4 | 1×2 1×3 1×3 | 1×5 1×4 1×4 | 1×5 1×5 1×6 | 1×3 1×4 | 56 |
| 2 | Number of stakeholders invited to fill in the questionnaire of the SCIROCCO tool | 10 | >20 | 9 | 11 | 12 | |
| | Number of individual questionnaires completed (target max. 10 local stakeholders per region) | 10 | 5 | 7 | 11 | 9 | 42 |
| | Number of participants in face-to-face workshop | 9 | 5 | 7 | 11 | 5 | 37 |
| 3 | | Study visit Basque Country (GP) | Study visit Norrbotten (dimension) | Study visit Puglia (GP) | Study visit Scotland (GP) | Study visit Scotland (dimension) | |
| | Number of actively involved participants in the study visit (transferring and receiving region) | 15 | 19 | 22 | 14 | 15 | 85 |
| | Number of actively involved participants from the receiving region in the study visit (max. 5 per region) | 5 (one left early) | 4 | 5 (Olomouc) 6 (Scotland) | 6 (Basque country) 4 (Puglia) | 4 | 34 |

GP, Good Practice; SCIROCCO, Scaling IntegRated Care in Context.

### Recruitment for self-assessment process (step 2)

All the regional SCIROCCO partners received instructions on how to perform the self-assessment in their regions and how to recruit the local stakeholders including predefined selection criteria. Despite the provided instructions and predefined invitation letter, the regional SCIROCCO partners needed to provide additional efforts to explain to the local stakeholders how to participate in the self-assessment process. As a result, extra informative documents and illustrative videos on how to use the SCIROCCO tool in the self-assessment process were provided. Furthermore, the tool was translated in three additional languages. In the regions, different procedures to identify the local experts in the self-assessment process were implemented. In four regions, the regional SCIROCCO partners recruited the local experts to participate in the self-assessment activities (regions 1, 2, 4 and 5). Another regional partner identified local stakeholders with the support of the local steering group (region 3). In addition, the recruitment in the regions varied in using different communication methods and in the length of the process. In some regions, there was a need

to clearly communicate the added value and benefits of the assessment process (regions 1 and 5 and organised a prior meeting to explain the self-assessment process. One region (region 4) translated the prepared documents. Another region called the identified experts by phone (region 2). In this region, some of the stakeholders expressed that it was hard to respond to the tasks (filling in the online tool) as the concept of IC is not an urgent topic on the agenda of their country. Subsequently, the recruitment of local stakeholders was a difficult and long procedure for this region. Finally, in some regions, it was a challenge for the local experts to reserve time to participate in the activities (regions 2 and 3).

### Recruitment for twinning and coaching (step 3)

The local experts who participated in the twinning and coaching activities were identified by the regional SCIROCCO partners in all five regions. The recruitment of experts varied per region, in terms of use of different communication channels and in length of the process. Exact details on how the experts were recruited could not be retrieved from the data. A few experts were already

**Table 6** Experiences of stakeholders about study visit (n=40) (figures in %)

| Question | Answer categories | | | | |
|---|---|---|---|---|---|
| | Very unclear | Unclear | Neither clear nor unclear | Clear | Very clear |
| Q1. Prior to the study visit, how clear was the information provided on the content and process of the study visit? | 5 | 0 | 5 | 55 | 35 |
| | Strongly disagree | Disagree | Undecided | Agree | Strongly agree |
| Q2. Were you able to ask and discuss everything you wanted during the study visit? | 0 | 7.5 | 10 | 35 | 47.5 |
| | Much less than expected | Less than expected | As expected | More than expected | Much more than expected |
| Q3. How well did the study visit matches your expectations? | 0 | 10 | 22.5 | 35 | 32.5 |
| | Not at all influence | Slightly influence | Somewhat influence | Moderately influence | Extremely influence |
| Q4. To what extent do you think the content discussed during the study visit should influence decisions in your region? (one answer missing) | 0 | 7.5 | 42.5 | 40.5 | 7.5 |

involved in previous activities within the SCIROCCO project, and most experts were new to the project.

### Participant responsiveness

The majority stakeholders were observed to have good rapport with the study visits. Most stakeholders who participated in the study visits indicated that the information provided on the content and process prior to the study visit was clear. Also, most stakeholders agreed that they could ask and discuss what they wanted during the visits and the visits matched with the expectations of most of them. Lastly, the majority of the experts indicated that the content of the study visit should somewhat or moderately influenced decisions in their region (see table 6 for all details).

Most stakeholders indicated appreciation for the usefulness of the study (explicitly addressed in 4 out of 5 study visits). The sharing of experiences and collaboration during the study visits were explicitly mentioned in two focus groups. A participant mentioned in focus group 4 (FG4) that '*we think you have something you can share with us and something we can learn from you*'. The study visits were considered as an inspiration and regarded as an '*injection of optimism*' (FG1) to improve elements in their own regions. In particular, the onsite experiences were appreciated in three study visits, were practices were visited '*in their real context*' (FG 2). However, in another study visit a respondent indicated that the visit was a bit less concrete and practical than expected.

The scheduled amount of time for the study visits was in some cases regarded as sufficient, and in other cases, this was perceived as insufficient. Some respondents indicated that they did not have enough time to reflect on information that was shared. Another respondent mentioned that prior to the visits, it would be useful to go through the tool and have a meeting or webinar with the transferring regions to have a better understanding of what will be the interest of the visiting regions. When two regions participated as receiving regions in the study visit, a few stakeholders indicated that more time would probably be needed for the study visit. On the site of the study visit organisers (transferring regions) it was indicated in two focus groups that the organisations of the visit might have benefited from more time to prepare to, for example, arrange '*even more practical examples*' (FG4) and to be able to '*to have the perfect team in place*' (FG3).

During the study visit, the tool was tested in facilitating the discussion between the regions. The use of the tool was regarded in four study visits as a support to facilitate the discussion by providing a structure to the conversation. Not all stakeholders had experience using the SCIROCCO tool in the study visits. Two respondents explicitly mentioned that they thought the tool was difficult to understand the first time. A difficulty experienced with the tool was mentioned in the language of the tool, as it was not regarded as the simplest English. According to a stakeholder, the terms used in the tool need to be locally interpreted, '*as it should be translated into a local terminology and context*' (FG3). Furthermore, it was indicated by a few stakeholders that having a translator present during the study visit was critical.

### Conditions to facilitate implementation
#### Project coordination, effective communication and guidelines
The central (strategy) coordination was regarded to be a supportive factor to implement the steps of the strategy

in the five regions (according to all partners). Since the start of the strategy, biweekly telecom conferences (virtual meetings) took place over the course of the entire implementation period to facilitate effective communication with the partners. During these telecoms, among other things, the draft methodologies designed for the scaling up steps were discussed. Several additional online meetings within the consortium were organised to make sure that all partners understood how to implement the different steps in their region. In addition, five project assembly meetings were organised where the partners were given the opportunity to share and discuss their work on implementing the activities in the five regions. Furthermore, written guidelines on how to implement the different activities in the regions were also shared by email to the regional partners. In addition, three sets of detailed guides were designed to guide the use of the online SCIROCCO tool. The full set of instructions was also included directly in the online version of SCIROCCO tool. In addition, several supportive training materials and demo videos were developed and could be accessed online. The available methodology was regarded to be helpful in providing consistency in the self-assessment processes and the twinning and coaching process.

### Flexible approach

The flexible approach within the strategy to develop the exact design of the steps of the strategy during the implementation and the opportunity to sometimes deviate from the original plan was regarded as a facilitator for implementation as indicated by the regional partners of two regions. One regional partner in interview 7 (I7) indicated: '*you give kind of an idea on how it should work, but then in real life things happen in a different way*'. One regional partner indicated that the flexible/open approach for developing the steps could have influenced the planning of the strategy, because in-depth discussions among the consortium partners or the representatives of regions sometimes took more time than anticipated.

### Supportive attitude of regional partners

The exact design of the steps of the strategy were developed in collaboration with the support of other work packages and the project coordination. The collaboration among the different regional and supportive partners was regarded as open and a positive experience which was explicitly mentioned by four partners. A facilitator for implementation mentioned by two regional partners was regarded in the commitment of all the project partners to achieve the objectives of the project. Also, one partner further indicated that the flexible attitude of all project partners was regarded to facilitate implementation.

### Regional conditions

The regional partners of two regions indicated that they had several co-workers from their region involved in the project that supported the implementation of the project activities in their region. For example, region 3 had set up

a local steering group, which appointed people to be part of a local project team, including people with different roles and responsibilities. This local working group worked on the activities of the SCIROCCO project, and from this group, some stakeholders were also recruited to participate as the stakeholder team throughout the project.

Lastly, before step 2 was implemented in all the five regions, the designed process was pilot tested in one region. One regional partner expressed that this was regarded as facilitative for the implementation in the other regions. For the implementation of the GP assessment (step 1), the regional partner of another region indicated that it perhaps would have been good to also pilot test the method in their own region before other regions followed.

### Context

The different regional contexts of the five SCIROCCO regions were found to influence the implementation of the steps in the five regions. Several context factors may have moderated adherence to the various scaling up steps.

### Language and conceptual adaptations on IC

During implementation, an unanticipated activity was the translation of the tool to improve stakeholders' experience with the tool. In addition, the provided invitation letters needed to be translated and all the individual assessments (for step 1 and 2) as well as consensus-building workshops were held in local languages except for one region, which was also held the workshop in English (region 3). During the implementation of the project, it was observed that the translation of the tool is not enough. To improve the understanding of the tool, it was indicated that the concepts of the tool needed to be adjusted to local linguistic and contextual aspects of IC. However, this cross-cultural adaptation was found not to be feasible in the duration of the strategy, and the recommendation was that this should be considered as a potential improvement of the tool in the future.

### Level of development in IC

The variation of the regions in the level of development and implementation of IC influenced the implementation. During the implementation of the SCIROCCO steps 2 and 3, the five regions were able to define the scope for the self-assessment process and the twinning and coaching activities to reflect the structure of their healthcare systems, the concept of IC (step 2) and the local needs and strategic priorities for IC (step 3). In one region, region 2, the concept of integrated of care was relatively new. For this region, IC is not widely known, and stakeholders have limited knowledge about nor experience with this approach. Hence, the assessment processes proved to be a complex task for the region, especially in the engagement with local stakeholders. Also, the region did not play the role of transferring region due to low maturity scoring across all SCIROCCO dimensions.

Furthermore, it was indicated by one partner that in the regions where the IC agenda was strongly established, the implementation of the activities of the project ran smoother.

## Local perspectives

The variation in implementing the steps may have also been influenced by the different perspectives of the SCIROCCO partners in the regions. One supportive partner mentioned that the regions varied in their perspectives as some were more policy driven, some more practical or research focused, which could have influenced the implementation, since the implementation of steps 2 and 3 were adapted to the regional context and needs.

## Changes in local environment

During implementation of SCIROCCO, changes in the local environment of the different participating regions were indicated to possibly influence implementation. Some organisational changes occurred in the public authorities where the regional partners were affiliated. For one region, the focus of their health system agenda changed. In another region, the change in management of their organisation influenced how well known the project was and how much leverage the project had. Another regional change during implementation was observed on the level of a GP. One GP was embedded as part of the routine practice at national level. As such, this GP could not be included in the second assessment on the GP in step 1.

## DISCUSSION

This study assessed the implementation fidelity of a step-based scaling up strategy for IC, which was implemented in five European regions as part of a cross-national project. The results show that all steps of the scaling up strategy were implemented with acceptable fidelity. All five regions conducted the GP self-assessments (step 1) and undertook the self-assessment process of their healthcare system context in their regions (step 2). Each region also participated in the twinning and coaching process and wrote one or two action plans (step 3). In addition, the targets that were set for each step were met or even exceeded (frequency). However, some deviation was also found. First, the GP assessments in the regions were implemented twice (step 1), and one region did not participate as transferring region in the twinning and coaching activity (step 3). Also, the duration of the steps 1 and 2 took longer than anticipated. Lastly, the coverage of the local experts per region who participated in the three scaling up steps varied in structure and size.

We found several factors that influenced implementation fidelity of the scaling up strategy. These included facilitating conditions in the flexible approach and coordination, positive participant responsiveness, regional variability in participant recruitment procedures and several contextual factors. Supportive influencing factors for implementation of the three steps in the five regions were found in good coordination to implement the steps, the use of effective communication strategies among the partners, the provision of guidelines on how to implement the processes and the SCIROCCO tool in the regions. Furthermore, the flexible approach, the supportive attitude of partners and certain regional conditions (ie, having a local steering group and regional pilot testing) have facilitated the implementation of the steps in the regions. A last important factor was the positive experience of the local stakeholders that participated in the study visits.

Several factors found in facilitating conditions (the flexible approach), the local variations and adaptions in participant recruitment procedures and several contextual factors moderated adherence to the implementation of the strategy. The contextual factors that influenced the implementation of the strategy were reflected in the local differences in language and conceptual understanding of IC among the regions. In addition, the different healthcare system contexts, level of development in IC and the local needs and strategic priorities resulted in variations in implementation of the strategy. Lastly, regional changes during the implementation of the strategy also influenced adherence. For example, for step 2, a selected GP could not be included in the second assessment as, during the lifespan of the strategy, it was embedded as part of the routine practice at the national level.

The implementation of the strategy consisted of a partly standardised and partly flexible approach. Flexible interventions present a challenge for evaluators, as flexibility and fidelity can be at odds with each other. The flexible approach to adapt or deviate from protocols is in contrast with examining the implementation fidelity of an intervention to its protocol. The discussions about the conflict on how to consider fidelity and the extent to which adaptation across contexts is fair or needed are unresolved.[25] However, adapting health service delivery to local conditions has several advantages. The variety in implementation found in our study corresponds to the recognition by Carroll et al[16] that an 'intervention cannot always be implemented fully in the real world. Local conditions may require it to be flexible and adaptable'.' Furthermore, it has the potential to improve programme effectiveness by means of increased programme feasibility and increased user adoption.[16] In such a case, it is important to take into account flexibility during implementation because of needed adaption to the local conditions. This occurred within SCIROCCO as the scope of the activities were informed by each individual region reflecting the structure of their healthcare systems, the framework of IC and the local need and strategic priorities for IC. To draw valid conclusions about the fidelity of an implementation, it is important to accomplish a good balance between adherence to a protocol on the one hand and the required flexibility for adaptations on the other hand.

To our knowledge, this study is the first to assess the fidelity of the implementation of an international scaling up strategy for IC. Most studies that assess the implementation fidelity focus on the implementation of national (complex health) interventions or programmes.[26–28] Consequently, we were not able to compare our findings to similar studies. Notwithstanding, our study shows that assessing the implementation of an international step-based scaling up strategy by using the modified version of the Conceptual Framework for Implementation Fidelity[17] is feasible. Hence, we encourage others who work or plan to work on similar strategies to study the implementation fidelity of such strategies to clarify the factors that are associated with the implementation of their strategy.

Nolte *et al*[29] explored the national experiences with developing, implementing and impacting the wider system context of IC in three countries. They concluded that 'a more formal strategy' for expansion is needed, as the wider dissemination of the projects studied occurred in 'an incremental and somewhat haphazard way'.[29] The scaling up strategy presented in this study focuses on the adoption of IC initiatives to other country settings, which is challenging because of different contexts. The scaling up strategy provides an encouraging example of a structured approach to scaling up of IC initiatives in an international context. The results of this study provide insight in the factors that need to be considered when implementing the strategy.

### Strengths and limitations
The opportunity to assess the implementation of the SCIROCCO strategy in its real-life setting of five European regions enabled us to collect several types of data, to triangulate data, and hence enhance in-depth insight into the implementation. Furthermore, our role within the project provided a unique opportunity to obtain a close collaboration with the partners to collect the data needed for the implementation fidelity assessment. However, being a partner in the consortium responsible for undertaking the evaluation activities, the subjective experiences of the participants (supportive partners, regional partners and local stakeholders) could have been influenced by social desirability and/or recall bias.

We were aware of our role, and to safeguard objectivity, we did not interfere with implementation of the project. However, due to time constraints and a high demand on the participants, we were not able to collect data on all preplanned moments that limited the study coverage of the potential factors influencing the implementation of complex interventions. Only data from one regional partner could be collected during implementation, whereas the rest of the data were collected at the end of the implementation. This limited our insight into the experiences and involvement of participants in other steps. To be able to explain ratings more in-depth, we recommend future researchers to examine participant experiences at several moments during the process of implementation. In addition, we have no specific information about the non-responders of the focus groups; since local stakeholders were recruited by the regional SCIROCCO partners we did not have insight in how many local stakeholders were approached by the partners. Therefore, we were unable to track the response rate in the study. Several factors may have contributed to non-responses, which could be the time demands on some respondents of participating in the SCIROCCO project itself (it took at least two full days and travelling time) or not seeing the immediate benefit from participating.

Due to the nature of the strategy, where the methodologies of the steps were developed during the implementation (including identification of the number of stakeholders), measurement of adherence was challenging. This corresponds to other studies who also explored the implementation fidelity with a lack of defined points of references for assessment.[26 27 30] The opportunity to work closely with programme implementers ensured us to be kept up to date on the changes regarding revisiting reference points with the evolution of the strategy.

### CONCLUSION
Overall, the SCIROCCO strategy was regarded an encouraging approach offering regions a tailored but flexible path intended to facilitate progress in IC. The multimethod design of this study has yielded knowledge about what elements are involved in implementing a European scaling up strategy concerning IC initiatives. The use of a theoretical framework helped us to document the process by which the strategy was implemented and understand the level of implementation fidelity achieved. The insights obtained could support other regions, not being part of the SCIROCCO project, on what to consider when interested to use the tool and processes to achieve knowledge transfer with other regions to ultimately scale up IC initiatives. Implementing the strategy might be challenging for regions that are less known to concept of IC and when the strategy gets modified depending on the local context of regions. Furthermore, the implementation of the steps in the five regions and this evaluation study took place within the boundaries of a project, where dedicated resources were provided. An important question is whether the SCIROCCO strategy will be as successful without these available resources and whether the evaluation study will encounter other factors that apply to the implementation in these settings. To test the external validity of study findings, it would be recommended to also test the SCIROCCO strategy in other countries both inside and outside Europe. Furthermore, lessons on implementation of the developed action plans would also be useful for other interested regions.

**Acknowledgements** We would like to thank all Scaling IntegRated Care in Context (SCIROCCO) project partners and local experts for their time and valuable contribution in the study.

**Contributors** LG and HJMV provided valuable contributions to the design of the work and provided substantial contributions to the analysis of the work. Both

authors provided significant contributions to the acquisition of data, interpretation of data and writing of the manuscript. INF and DD reviewed the manuscript critically and substantively revised the work. All authors read and approved the final manuscript.

**Funding** This study was undertaken within the SCIROCCO project, which was cofunded by the Health Programme of the European Union under Grant Agreement 710033 (CHAFEA).

**Disclaimer** The funding body was neither involved in the design of the study nor the collection, analysis, and interpretation of data and writing of the manuscript.

**Competing interests** None declared.

**Patient and public involvement** Patients and/or the public were not involved in the design, or conduct, or reporting, or dissemination plans of this research.

**Patient consent for publication** Not required.

**Ethics approval** The conduct of this study was granted permission by the Institutional Review Board of the Vrije Universiteit Brussel (reference: B.U.N. 143201734384). Participant consent was requested before the interviews and focus groups in the form of an informed consent. Signed written consent forms were completed by all participants.

**Provenance and peer review** Not commissioned; externally peer reviewed.

**Data availability statement** The datasets used and analysed during the current study are available from the corresponding author on reasonable request.

**Open access** This is an open access article distributed in accordance with the Creative Commons Attribution 4.0 Unported (CC BY 4.0) license, which permits others to copy, redistribute, remix, transform and build upon this work for any purpose, provided the original work is properly cited, a link to the licence is given, and indication of whether changes were made. See: https://creativecommons.org/licenses/by/4.0/.

**ORCID iDs**
Liset Grooten http://orcid.org/0000-0003-4515-2596
Dirk Devroey http://orcid.org/0000-0002-6083-2998

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
