## [Reviewer comments · BMJ Open]

ARTICLE DETAILS

TITLE (PROVISIONAL)	Assessment of the implementation fidelity of a strategy to scale-up integrated care in five European regions: a multi-method study
AUTHORS	Grooten, Liset; Fabbriotti, Isabelle Natalina; Devroey, Dirk; Vrijhoef, Hubertus J.M.

VERSION 1 – REVIEW

REVIEWER	Elizabeth Irungu Jomo Kenyatta University of Agriculture and Technology
REVIEW RETURNED	11-Nov-2019

GENERAL COMMENTS	Well written, detailed paper to describe the extent to which the SCIROCCO strategy was implemented as planned.  1. The authors make reference to the SCIROCCO tool numerous tools. It would be of great value if they would avail this tool for the readers to see. It can be included as an appendix. 2. The first step of the strategy is identification of 30 adoptable GPs and then assess their viability to remain with 15 GPs. The authors state that the assessment is done using a from developed in implementation. Provide more detail of the contents of the from. What are the specific details examined to assess viability? Page 4, line 17 -21. 3. Similar to the item above, please provide some more detail on what specifics are examined to determine maturity requirements of GPs and/or health system. Page 4, line 21-36 4. One twinning and one coaching activity per region was envisaged. Please explain how and who determined which region would be twinned with which and which would coach which. Page 5, line 3-5. For instance, is it pre-specified based on scores observed at steps 1 and 2? 5. Table 3 (page 8) – comes before the description of what twinning and coaching is on page 10, making it difficult to understand. You may include a footnote explaining these terms. In addition, the words transferring and coaching and adopting and receiving are used interchangeably. Be consistent. 6. Page 11, line 17 – the authors state that the expectation was that regions would upload action plans. We are not informed whether this was done by any or all regions and whether this is also part of the fidelity assessment. 7. A few typos: Page 9, line 55 – good Page 11, line 17 – intent Page 12, line 34 – dependent Page 16, lines 9,10 – SCIROCCO
---

REVIEWER	Jennifer Mann James Cook University, Australia
REVIEW RETURNED	02-Dec-2019

GENERAL COMMENTS	This work is an important addition to integrated care literature, most importantly, the way in which these models are best implemented and scaled up. The methodology is comprehensive and sound. The discussion is evidenced and provides exploration of the impact of regional differences on fidelity of a national scale-up. This is a very long manuscript and this takes away from the key messages at times; the format is difficult to follow and a review of the methodology and results sections for improved readability is recommended.
---

REVIEWER	Derek Tracy King's College London
REVIEW RETURNED	31-Dec-2019

GENERAL COMMENTS	Thank you for asking me to review this paper; the issue of more and better integrated care is an international one, but the area is highly complex and marked by a lack of consensus on what 'integration' means, or how it can/should be implemented. This paper reports on a testing, via qualitative methods, of a scaling up strategy of integrated care initiatives across five European regions. The authors found that the three main steps of scaling up were implemented with adequate fidelity; there were variations in expert participation. Factors influencing implementation included facilitation conditions and participant responsiveness. The area is important, and this paper is helpful to international audiences. I enjoyed reading it, and am pleased to see more of this type of work emerging – it is needed as there is currently a lack of research on the topic. I think the paper will be of value to a wide audience, and I look forward to seeing it published. My main comments to the authors are as follows: Introduction  - I believe that as well as challenges in measuring which factors contribute to success/failure in integrated care, there is an earlier issue of defining what one means by it. There are innumerable models incorporating various aspects of primary/secondary/tertiary physical and mental health in single or multiple age groups, and the involvement of various aspects of social care and/or governmental/commissioning agents. This is somewhat inevitable, but it makes even initial conversations on the topic problematic (what is 'your' integration compared with 'mine' – before we even get to conversations on demographics). This makes interpretation of 'what works' very difficult, even if we have such data. I think the authors need to tersely note this, and that we lack a clear language/agreed matrix of what 'the integrations' might variously look like; we previously wrote about some of these challenges - Tracy et al, Integrated care in mental health: next steps after the NHS Long Term Plan. British Journal of Psychiatry, 2019; 214:315-317. doi: 10.1192/bjp.2019.46 - Typo at the end of paragraph 2: should read "and Scotland" - "Step 1"; I found the choice of the word 'inspiring' odd as it feels very subjective. I suggest remove to just acknowledge that such sites had some preliminary evidence of success in implementing/setting up such innovations.
--

	- I found it all a bit theoretical without some description of at least one site. I recognise they will all be different, but is there room for a brief box outlining one site, what is interesting/novel about its integrating model? Otherwise this is hard for the reader to understand how this might relate to their own area/practice. - End of page 7, I did not understand the relevance of noting that the three facilitators were women; clarify or remove. The methodology is appropriate and results described adequately Discussion - “differences in language and conceptual understanding of integrated care...” also resonates with my earlier point that even in a given regional health care system, there might further be a rationale for quite different models. I am working on integrated care in South East London, and the needs and model are quite different from those in South West, and certainly North, London; this is further impacted by the priorities and aspirations of clinicians, managers, and clinicians. It goes back to a lack of language on the different types of integration. In a perfectly funded, resource limitless environment, we might wish to do integration differently in different areas, and of course we have real world restraints. The authors mention this in the following line or two but I am minded that this is more profound than is covered in the current iteration.
--	--

VERSION 1 – AUTHOR RESPONSE

Comments reviewer 1

1. The authors make reference to the SCIROCCO tool numerous tools. It would be of great value if they would avail this tool for the readers to see. It can be included as an appendix.

Authors response to the reviewers comments/suggestions: Thank you for the suggestion. We included the tool as an Appendix A to the manuscript.

2. The first step of the strategy is identification of 30 adoptable GPs and then assess their viability to remain with 15 GPs. The authors state that the assessment is done using a from developed in implementation. Provide more detail of the contents of the from. What are the specific details examined to assess viability? Page 4, line 17 -21.

Authors response to the reviewer’s comment/suggestion: We included the details on the assessment of the viability on page 10, below the result section (as the criteria were developed during the implementation): In order to evaluate the viability for scaling-up of the SCIROCCO GPs, a six-criterion assessment framework was built based on the criteria used in the European Innovation Partnership on Active and Healthy Ageing repository. The six criteria included: 1. Time needed for the practice to be deployed; 2. Investment per citizen/service user/patient (referring to marginal cost over previous situation);3. Evidence behind the practice;4. Maturity of the practice;5. Estimated time of impact of the practice and 6. Level of transferability of the practice.

3. Similar to the item above, please provide some more detail on what specifics are examined to determine maturity requirements of GPs and/or health system. Page 4, line 21-36

Authors response to the reviewer’s comment/suggestion: we included a description on the SCIROCCO tool in Box 1 on page 4 in the manuscript. We further provided all details of the tool in the added appendix A as suggested by the reviewer at point 1. We believe that these details are sufficient

to better understand the assessments of the maturity requirements.

4. One twinning and one coaching activity per region was envisaged. Please explain how and who determined which region would be twinned with which and which would coach which. Page 5, line 3-5. For instance, is it pre-specified based on scores observed at steps 1 and 2

Authors response to the reviewer's comment/suggestion: On page 5 we describe that: "The last step consists of the development of the process for the twinning and coaching activities of five European regions to support the creation of twinning and coaching relationships by using the information (steps 1 and 2 of SCIROCCO's strategy) to construct these relationships." The information in step 1 concerns the information on the maturity requirements of the prioritised 15 GPs which are assessed using the SCIROCCO tool. And the information in step 2 the five participating regions assessed their maturity in the adoption for integrated care of their region, using the online SCIROCCO tool, to identify strengths, gaps and areas for improvement. The information of step 1 and step 2 is used as information to help regions determine which region would be twinned with which and which would coach which.

5. Table 3 (page 8) – comes before the description of what twinning and coaching is on page 10, making it difficult to understand. You may include a footnote explaining these terms. In addition, the words transferring and coaching and adopting and receiving are used interchangeably. Be consistent.

Authors response to the reviewer's comment/suggestion: We included a footnote to the table explaining the terms transferring and receiving region to Table 3. Furthermore, we replaced all the adopting regions to receiving regions to be consistent; please see the track changes in the manuscript.

6. Page 11, line 17 – the authors state that the expectation was that regions would upload action plans. We are not informed whether this was done by any or all regions and whether this is also part of the fidelity assessment.

Authors response to the reviewer's comment/suggestion: Unfortunately, we were not able to collect data on this matter, and we also included this to the last sentence of the specific paragraph. On Page 11 we describe that: "no specific details could be retrieved on the last two intended activities." Were we refer to the previous sentences: All five regions participated as receiving regions and wrote an action plan reflecting the outcomes of the twinning and coaching activities. Once completed, the intend was to upload the action plans in the online SCIROCCO tool and share these with all relevant stakeholders. In the last step, the plan was to promote the outcomes of the knowledge transfer activities locally and across the regions.

7. A few typos:

Page 9, line 55 – good Authors response to the reviewer's comment: Thank you, we eliminated the typo.

Page 11, line 17 – intent Authors response to the reviewer's comment: Thank you, we eliminated the typo.

Page 12, line 34 – dependent Authors response to the reviewer's comment: Thank you, we eliminated the typo.

Page 16, lines 9,10 – SCIROCCO Authors response to the reviewer's comment: Thank you, we eliminated the typo.

Comments reviewer 2

This work is an important addition to integrated care literature, most importantly, the way in which these models are best implemented and scaled up. The methodology is comprehensive and sound.

The discussion is evidenced and provides exploration of the impact of regional differences on fidelity of a national scale-up. This is a very long manuscript and this takes away from the key messages at times; the format is difficult to follow and a review of the methodology and results sections for improved readability is recommended.

Authors response to the reviewer's comments/suggestions: Thank you. We included some adjustments to the text in the methods section and in the result section to improve the readability of the manuscript. Please see the track changes in these sections.

Comments reviewer 3

Introduction

- I believe that as well as challenges in measuring which factors contribute to success/failure in integrated care, there is an earlier issue of defining what one means by it. There are innumerable models incorporating various aspects of primary/secondary/tertiary physical and mental health in single or multiple age groups, and the involvement of various aspects of social care and/or governmental/commissioning agents. This is somewhat inevitable, but it makes even initial conversations on the topic problematic (what is 'your' integration compared with 'mine' – before we even get to conversations on demographics). This makes interpretation of 'what works' very difficult, even if we have such data. I think the authors need to tersely note this, and that we lack a clear language/agreed matrix of what 'the integrations' might variously look like; we previously wrote about some of these challenges - Tracy et al, Integrated care in mental health: next steps after the NHS Long Term Plan. British Journal of Psychiatry, 2019; 214:315-317. doi: 10.1192/bjp.2019.46

Authors response to the reviewer's comment/suggestion: In this study, we are trying to identify the determining factors in order to help clarify this discussion on the interpretation of 'what works.' We included the next text to the introduction in order to deepen this, on page 3: "Furthermore, in the presence of innumerable models of IC, there is no universal conceptual understanding of integrated care, which leads to a lack of understanding on what 'integration' might variously look like."

- Typo at the end of paragraph 2: should read "and Scotland"

Authors response to the reviewer's comment: Thank you, we eliminated the typo.

- "Step 1"; I found the choice of the word 'inspiring' odd as it feels very subjective. I suggest remove to just acknowledge that such sites had some preliminary evidence of success in implementing/setting up such innovations.

Authors response to the reviewer's comment: Point well taken, we removed the word inspiring.

- I found it all a bit theoretical without some description of at least one site. I recognise they will all be different, but is there room for a brief box outlining one site, what is interesting/novel about its integrating model? Otherwise this is hard for the reader to understand how this might relate to their own area/practice.

Authors response to the reviewer's comment: Point well taken. We included a description of one site in Box 2 to the manuscript, furthermore we refer to another article where all the sites are outlined on page 11: "A description of the GP of one site involved as transferring region in the twinning and coaching activity is provided in Box 2. Descriptions of all 5 twinning and coaching sites are outlined elsewhere [24]."

- End of page 7, I did not understand the relevance of noting that the three facilitators were women; clarify or remove.

Authors response to the reviewer's comment: This is an answer to item no. 4 on the COREQ (Consolidated criteria for REporting Qualitative research) Checklist: Was the researcher male or female?

- The methodology is appropriate and results described adequately.

Authors response to the reviewer's comment: Thank you very much.

Discussion

- "differences in language and conceptual understanding of integrated care..." also resonates with my earlier point that even in a given regional health care system, there might further be a rationale for quite different models. I am working on integrated care in South East London, and the needs and model are quite different from those in South West, and certainly North, London; this is further impacted by the priorities and aspirations of clinicians, managers, and clinicians. It goes back to a lack of language on the different types of integration. In a perfectly funded, resource limitless environment, we might wish to do integration differently in different areas, and of course we have real world restraints. The authors mention this in the following line or two but I am minded that this is more profound than is covered in the current iteration.

Authors response to the reviewer's comment: We are pleased to see this point resonates with the experiences of the reviewer. We deleted 'perspectives of the SCIROCCO partners in the regions' to overcome the suggested iteration.

VERSION 2 – REVIEW

REVIEWER	Elizabeth Irungu Kenya Medical Research Institute
REVIEW RETURNED	14-Feb-2020
GENERAL COMMENTS	The authors have adequately addressed the reviewer comments.
REVIEWER	Derek Tracy King's College London
REVIEW RETURNED	12-Feb-2020
GENERAL COMMENTS	My queries have been adequately addressed.